# A Scoping Review of Educational and Training Interventions on Parkinson’s Disease for Staff in Care Home Settings

**DOI:** 10.3390/nursrep15010020

**Published:** 2025-01-13

**Authors:** Stacey Finlay, Tara Anderson, Elizabeth Henderson, Christine Brown Wilson, Patrick Stark, Gillian Carter, Matthew Rodger, Mihalis Doumas, Emma O’Shea, Laura Creighton, Stephanie Craig, Sophie Crooks, Arnelle Gillis, Gary Mitchell

**Affiliations:** 1School of Nursing and Midwifery, Queen’s University Belfast, Belfast BT79 7BL, UK; sfinlay15@qub.ac.uk (S.F.); tanderson@qub.ac.uk (T.A.); elizabeth.henderson@qub.ac.uk (E.H.); c.brownwilson@qub.ac.uk (C.B.W.); p.stark@qub.ac.uk (P.S.); g.carter@qub.ac.uk (G.C.); laura.creighton@qub.ac.uk (L.C.); s.craig@qub.ac.uk (S.C.); scrooks08@qub.ac.uk (S.C.); agillis03@qub.ac.uk (A.G.); 2School of Psychology, Queen’s University Belfast, Belfast BT79 7BL, UK; m.rodger@qub.ac.uk (M.R.); m.doumas@qub.ac.uk (M.D.); 3Centre for Gerontology and Rehabilitation, School of Medicine, University College Cork, T12 YN60 Cork, Ireland; emma.oshea@ucc.ie

**Keywords:** Parkinson’s disease, care home, nursing home, long-term care, nursing, education, training, review

## Abstract

Background/Objectives: Parkinson’s disease (PD) is a complex neurodegenerative disorder that presents significant challenges for care home residents and staff. This scoping review aimed to synthesize evidence on PD education and training available to care home staff, examine existing programs and their effectiveness, and identify gaps in current educational approaches. Methods: A scoping review (ScR) was conducted and guided by the Preferred Reporting Items for Systematic Reviews and Meta-analysis extension for ScR (PRISMA-ScR) checklist. A comprehensive search of six electronic databases was conducted in September 2024. Studies focusing on PD education and training for care home staff were included. Data extraction and quality appraisal were performed, followed by thematic analysis to identify key patterns and themes. Results: Seven studies met the inclusion criteria. The thematic analysis revealed four main themes: improvements in PD knowledge and confidence, improvements in care practices and outcomes, the need for increased specialist education, and the incorporation of communication training. Educational interventions led to significant improvements in staff knowledge, confidence, and care practices. However, these studies also highlighted a critical need for more specialized PD training among care home staff. Conclusions: This review provides evidence of promise regarding the potential impact of PD-specific education on care home staff knowledge and practices. Future research should focus on developing and evaluating comprehensive, tailored educational programs to enhance the quality of care for people with PD in care home settings.

## 1. Introduction

Parkinson’s disease (PD) is a chronic, progressive neurodegenerative disease which encompasses a range of complex and fluctuating motor and non-motor syndromes, including tremor, rigidity, and bradykinesia [1]. PD is the second most common and fastest-growing neurodegenerative disease worldwide [2]. The World Health Organization (WHO) estimated that in 2019, approximately 8.9 million people were living with PD globally, a 200% increase over the preceding 25 years [3].

The prevalence of PD increases with age, and rates almost double every five years between the ages of 50 and 69 [4]. This is due to age-related changes, which lead to a loss of neurons and impair their ability to respond to stressors, increasing the likelihood of neuronal dysfunction and loss [5]. People with PD over the age of 65 often live in long-term care facilities [6,7]. Internationally, the prevalence of PD in care homes has been estimated between 5 and 48% [8,9]. Variations in population demographics, diagnostic criteria, and care facility practices may account for this wide range in prevalence rates.

The complexity of PD presents many challenges for care home residents and staff due to the unique pharmacological regimens and symptomatology spectrum associated with PD [10]. The optimal management of PD requires input from multiple healthcare disciplines due to the range of symptomatology characteristics of the disease [11]. However, care home staff are often the primary caregivers for residents with PD, as people living with this condition over the age of 65 are likely to live in long-term care facilities [6]. Care home staff are therefore responsible for managing the daily needs, medication regimens, and quality of life of people living with PD. It is essential that these staff have sufficient knowledge of PD to ensure the safety of residents living with this condition and provide optimal management of their symptoms [12].

A lack of education and training for care home staff regarding PD may lead to suboptimal outcomes for residents, such as a lack of stimulation, social isolation, and a loss of independence [10]. Education is especially important in this area as the challenges presented by the complexity of PD may be exacerbated by high staff turnover rates and staff shortages within care homes, which can lead to gaps in knowledge and inconsistencies in care delivery [13]. Previous educational interventions have shown promise within other settings. For example, an interprofessional education program in PD was found to elicit increased knowledge of PD, improve team attitudes, and improve practices among care staff from outpatient, acute care, and home care settings [14]. In addition, nursing students showed increased PD awareness following engagement with an audio podcast promoting PD awareness [15]. There exists a need to explore such educational initiatives and their impact within care home settings.

The present scoping review (ScR) aims to synthesize the evidence relating to PD education and training that is available to care home staff. A scoping review methodology was selected as it was well suited to the objectives of mapping evidence, exploring existing programs, and identifying gaps in PD education in care homes. The broad nature of the research question required a method capable of encompassing diverse study designs, outcomes, and approaches within the available literature. The anticipated heterogeneity in the evidence base further supported the choice of a scoping review over a systematic review. Additionally, the exploratory aim to identify gaps and guide future research aligned with the scoping review methodology. This ScR sought to address the following objectives:(1)Identify the existing literature on the education and training programs available for nursing and nursing assistant staff in care homes regarding PD.(2)Analyze the content and effectiveness of the PD education and training programs available to nursing and care staff in care home settings.(3)Propose recommendations for enhancing the quality and accessibility of education and training opportunities for nursing and care staff in care homes to improve the care of people living with PD in care homes.

## 2. Materials and Methods

### 2.1. Design

An ScR was conducted following the methodology proposed by Arksey and O’Malley (2005) [16], guided by the Preferred Reporting Items for Systematic Reviews and Meta-analysis extension for ScRs (PRISMA-ScR) checklist [17]. This process involved establishing a research question, identifying relevant studies through the development of eligibility criteria, and selecting relevant studies. Data from the relevant studies were then extracted and analyzed, and the results were summarized and reported. A scoping review protocol was retrospectively registered at https://doi.org/10.17605/OSF.IO/NSFA5 (accessed on 9 January 2025). Please see Appendix A for a copy of the PRISMA-ScR checklist.

### 2.2. Search Strategy

The search strategy was developed with the review team, including input from a subject librarian. A preliminary search was also completed using Google Scholar. The PEO (Population, Exposure, Outcome) framework was used to aid the development of the research question and help devise eligibility criteria for relevant studies [18,19]. Therefore, to address the question of what education or training is available to registered nursing and nursing assistant staff working in care home settings, the search terms related to (P) care homes; (E) Parkinson’s disease; and (O) training or education.

Six electronic databases were searched in January 2024: CINAHL Plus (EBSCOhost), PsycINFO (OVID), Medline All (OVID), Scopus (Shibboleth), Embase (Shibboleth), and PubMed. The search terms listed in Table 1 were used, with MeSH terms also used where available in each of the databases. The EMBASE search strategy is provided as an example in Appendix A.

### 2.3. Eligibility Criteria

No restrictions were placed on the date of publication. All types of empirical studies were included. Studies not in the English language were excluded. The included population were care home staff who provide direct care to residents living with PD; this included nurses and nursing assistants. Table 2 presents the full details of the inclusion and exclusion criteria used in this ScR.

### 2.4. Data Extraction

Covidence software (https://www.covidence.org/ accessed on 9 January 2025) was used to aid in the removal of duplicates, the screening of the literature, and data extraction. Title and abstract screening were conducted independently by SF and GM, with both reviewers screening 100% of the records. Conflicts at this stage were resolved through discussion with EH. Full-text screening was then conducted independently by SF and GM, with both reviewers again screening 100% of the articles. Conflicts at the full-text stage were resolved through discussion with SC. For data extraction, a standardized extraction template was utilized within Covidence to ensure consistency across all studies. Key data points, including the study design, population characteristics, intervention details, outcomes, and key findings, were extracted systematically. Both reviewers (SF and GM) independently performed data extraction for all included studies, with regular cross-checking to ensure accuracy. Any discrepancies in data extraction were discussed and resolved collaboratively to maintain reliability. Although not mandatory for ScR [17], a quality appraisal of the included studies was conducted to provide readers with an understanding of the quality of the evidence. The JBI checklists for quasi-experimental studies and qualitative research [20] were used for the quality appraisal of the included studies.

### 2.5. Data Analysis

This scoping review employed thematic analysis as a systematic approach to identifying, analyzing, and interpreting patterns within the data [21]. The analysis followed a six-step framework as outlined by Braun and Clarke [22]. First, data familiarization was undertaken, where SF reviewed the extracted data multiple times to ensure a deep understanding of the content. This step involved reading and re-reading the data, making initial notes, and identifying potential areas of interest. Next, initial codes were generated through an inductive process, systematically labeling features of the data relevant to the research objectives. Codes were applied across the entire dataset to ensure comprehensive coverage. Following this, themes were searched by collating similar codes into broader categories that reflected meaningful patterns or concepts within the data. In the stage of reviewing themes, these initial themes were critically evaluated to ensure coherence and alignment with the dataset. This involved refining themes by merging overlapping ones, discarding those that were not well supported, and verifying their relevance to the research question. During the step of defining and naming themes, the essence of each theme was clarified, with definitions established to distinguish them. Finally, the reporting of themes involved organizing the results into a coherent narrative supported by illustrative examples from the data. This iterative process was led by SF, with ongoing re-checking, guidance, and feedback provided by the review team to ensure rigor and reliability.

## 3. Results

A total of 10,405 studies were initially imported for screening. After removing 162 duplicates, 10,243 studies proceeded to title and abstract screening. Of these, 10,037 were deemed irrelevant and excluded. Full texts were sought for the remaining 206 studies; however, 29 studies could not be retrieved due to reasons such as inaccessible or outdated journal platforms, paywalls, or the studies not being available in any institutional or public database. Overall, 177 full-text studies were screened. Following this process, seven studies met the inclusion criteria and were included in the review, as illustrated in the PRISMA-ScR flowchart in Figure 1.

### 3.1. Characteristics of Included Studies

Five studies used a quantitative approach, employing questionnaires [23,24,25,26,27]. One study utilized quantitative methods analyzing observations of participant interactions and individual discussion and feedback sessions between participants and researchers [28]. Finally, one study used a mixed-methods approach, combining participant questionnaires and interviews [29]. Further details of the study characteristics are provided within Appendix A.

All included studies were conducted in care home settings, either exclusively or partially. In the case of the one study that included both acute care wards and residential aged care facilities [24], only the data relating to residential aged care facilities were included in this review.

Three included studies were conducted in England [23,26,29]. Two were conducted in Australia [24,25]. One study was conducted in Sweden [28] and another in the United States of America [27]. Further details regarding sample size and participant demographics are presented within Table 3.

### 3.2. Quality Appraisal

The quality of the literature included in this scoping review was appraised using the JBI critical appraisal checklist for quasi-experimental studies. The JBI checklist for quasi-experimental studies is a recognized tool for the critical appraisal of quality and risk of bias [20]. Details of the quality appraisal of each study are included in Table 4 below, and individual appraisal forms are available upon request. The JBI checklist for quasi-experimental studies does not include a scoring system to quantify the quality of studies. However, to assist with quantifying the quality of the included studies, a scoring system out of 9, based on the nine items on the checklist, was implemented.

Studies scoring between 0 and 4 were classified as low quality, those scoring 5–7 as moderate quality, and those scoring 8–9 as high quality. None of the included studies were considered high quality, while one included study was classified as low-quality following appraisal [29]. The remaining six studies were classified as moderate quality. In all studies, the participants included were similar. However, none of the researchers provided information about whether participants were receiving similar education or training other than the exposure or intervention limiting the possibility of establishing if there were any confounding factors that may have influenced the results.

All studies aside from Oates et al. (2016) [26] measured outcomes both pre- and post-intervention. Five out of seven studies completed a follow-up with the participants [23,24,25,27,28]. In studies that did conduct a follow-up, there was consistency in how outcomes were measured in the follow-up. All studies aside from Coles et al. (1995) [29] measured outcomes in a reliable way; however, not all studies provided details on the use of appropriate statistical analyses.

### 3.3. Study Results

Following the thematic analysis, the following four themes were developed from the data: (1) improvements in PD knowledge and confidence; (2) improvements in care practice and outcomes; (3) lack of specialist PD education; and (4) the incorporation of communication training.

#### 3.3.1. Theme One: Improvements in PD Knowledge and Confidence

Theme one highlights the substantial impact of educational interventions on enhancing care home staff knowledge about PD. The reviewed studies reveal several key patterns in how these interventions affect staff understanding and practices.

Educational interventions lead to significant improvements in knowledge both immediately and over time. A medication management education program resulted in increased knowledge among nurses, with perceived and actual knowledge levels rising from pre-test scores of 68% to 85% post-intervention, and these gains were maintained at follow-up [24]. Similarly, another study found sustained improvements in staff knowledge from a baseline score of 60% to 75% post-intervention [25].

In one study, those who received training reported a higher average confidence level in caring for people with PD of 7.7 compared to 5.9 in those who had not received any PD training [26]. In the same study, respondents felt further training would be beneficial. In addition, a communication-focused training program was highly rated by all participants, with 100% of respondents finding the training useful [29].

It was also evident that targeted educational programs addressed specific knowledge gaps. An intervention on non-motor symptoms led to an increase in knowledge from a baseline of 0% awareness to an average improvement of 40% [27]. A comprehensive care home education program, which included a PD-specific module, showed an overall increase in knowledge from a baseline score of 50% to 70% post-intervention, although specific improvements related to PD were not detailed [23].

#### 3.3.2. Theme Two: Improvements in Care Practices and Outcomes

Training programs led to measurable improvements in care outcomes. For instance, a fall prevention educational intervention in care homes resulted in a decrease in the rate of falls by approximately 9% per month over the course of the 12 months following the intervention [25]. Another study reported that 64% of individuals with PD experienced negative outcomes in respite or long-term care settings due to inadequate specialist training of care home staff, but positive patient experiences were frequently associated with staff who had received thorough PD training in the form of improved quality of personalized PD care and greater adherence to medications [23].

Educational programs that focused on PD medication were commonly reported within this review [23,24,25]. In two Australian studies [24,25], education about PD medications led to a variety of improvements in practice, including increased adherence to medication schedules, better understanding of potential side effects, and improved management of complex medication regimens. One study reported that their Parkinson’s Disease Medication Protocol Program significantly increased nurses’ knowledge and confidence in managing PD medications, leading to more accurate medication administration and reduced errors [24]. Further, another study demonstrated that staff training on medication management contributed to improved resident outcomes, such as better symptom control and reduced hospital admissions related to medication issues. These findings emphasize the importance of targeted medication education in enhancing both staff competency and resident care quality [25].

In addition to reduced falls, improvements in personalized care, and medication adherence, education about PD can also support the development of communication practices among nurses working within care homes [28]. In this Swedish study, a communication partner training program between nurses and residents with PD led to perceived improvements in the quality of communication post-intervention from both staff and residents.

#### 3.3.3. Theme Three: Lack of Specialist PD Education

A recurring theme across the included studies was the lack of specialist PD education for care home staff. The included studies highlighted the low prevalence of previous PD-specific training among care home staff, with studies reporting only 50% [29] and 59.6% [26] of care home staff having received prior PD-related training. Another study discovered that only 6.1% of aged care facility staff had received training or education specifically related to PD and its pharmacological management [24].

Significant gaps in nurses’ knowledge regarding the management of PD, particularly in the administration of PD medications, were also identified [24]. This study revealed that without specific clinical education, nurses often lack the necessary expertise to administer PD medications safely and effectively. The absence of such knowledge can result in serious negative outcomes, such as delayed medication administration, dose omissions, or medication errors, which can adversely affect patient health and well-being [24].

Another study highlighted a significant gap in specialist training, specifically in the recognition and management of non-motor symptoms in individuals with PD [27]. Despite the profound impact these non-motor symptoms can have on the quality of life for individuals with PD, none of the nurses in this study demonstrated awareness of non-motor symptoms prior to the educational intervention.

The included studies [24,26,27,29] also provided recommendations about developing specialist PD education in care homes. First, future educational programs should ensure that staff are well versed in the full spectrum of PD symptoms, including both motor and non-motor symptoms. Non-motor symptoms, which are often under-recognized and under-treated, require particular attention [27]. Additionally, specialized education on medication management is critical, as safe and effective medication practices are essential for preventing errors and improving patient outcomes [24]. The studies also emphasize the importance of ongoing training, with regular refresher courses and updates integrated into professional development to ensure staff stay informed about the latest advances in PD care and retain essential knowledge [24,26]. Furthermore, expanding access to specialized PD training programs is vital, especially given the low percentage of staff who have received prior PD education. This could include mandatory training for all care home staff, ensuring that educational resources are widely available and accessible to improve overall care quality [23,26].

#### 3.3.4. Theme Four: The Incorporation of Communication Training

Communication was consistently identified as a critical component of PD care in educational interventions. Evidence from several studies demonstrated the benefits of communication training for care home staff, though the progressive nature of PD often limited the sustainability of these improvements. These programs consistently highlighted the vital role of communication skills in effective care, even when not specifically tailored to the distinct challenges posed by PD.

One study found that training care home staff as communication partners enhanced interactions with residents living with PD, though the long-term benefits were diminished due to the degenerative progression of the disease [28]. This highlights the necessity of communication strategies that adapt to the changing needs of individuals with PD.

Another study incorporated communication training into an educational program that included a module on PD, highlighting the importance of communication in delivering quality care [29]. Similarly, a different program integrated modules on communication and PD, emphasizing the relevance of these skills in care settings [23]. However, neither program focused specifically on the unique communication challenges experienced by individuals with PD, suggesting an area for further development [23,29].

The use of communication partner training, a method previously validated for individuals with aphasia, was evaluated in a study involving a nurse and a PD resident dyad [28]. While initial improvements in communication were observed, the effects were not sustained due to the resident’s progressive cognitive and motor decline.

Notably, two studies in this review developed their training programs broadly for care staff working with older adults rather than exclusively targeting PD-specific care [23,29]. These programs included general communication modules, which offered some indirect benefits for PD care. However, the programs did not address the distinct and progressive communication challenges faced by people with PD, highlighting the need for more specialized training.

## 4. Discussion

This ScR aimed to synthesize evidence relating to PD education and training that is available to care home staff. The included literature highlights the potential for educational interventions for improving care home staff knowledge, confidence, and various aspects of care. Educational interventions may address gaps in staff understanding of motor and non-motor symptoms, medication management, and communication challenges. However, the sustainability of these improvements depends on ongoing training and support.

The training and education interventions evaluated within the included studies may lead to an improvement or positive impact on care practices. This reflects previous research in which improved patient outcomes were observed when people living with PD were referred to health professionals with specialist knowledge of their condition [30]. Therefore, providing those who provide the majority of care to those living with PD in care homes, the care home staff, with increased specialist knowledge is likely to improve care home resident outcomes. In addition, some participants reported increased interest in improving their practices in relation to PD following training [24]. This highlights the positive impact training programs may have for not only increasing knowledge but also promoting increased professional development and continued learning.

In the reviewed studies, communication was highlighted as an important aspect of training [23,28,29]. Reduced communication between health professionals and patients can result in decreased patient autonomy and increased risk of poor outcomes [31,32]. This may be heightened by the communication difficulties associated with PD, which have been shown to have a substantial impact on quality of life [33]. Improving the communication between healthcare professionals and people living with PD may facilitate a positive impact on their care by reducing these factors. Two of the included studies designed their training programs for care staff working with older adults in general [23,29]. These studies did not address the specific communication challenges of PD, suggesting that communication training for older adults in general may indirectly benefit those with PD. However, the distinct nature of PD communication impairments presents a strong case for developing tailored, PD-specific, communication training.

In one study, the findings related to communication partner training were not sustained, which the authors attributed to the progressive nature of PD [28]. Communication partner training has shown variable results in other areas such as oncology settings [34], and so further research is necessary to evaluate the benefits of such training. Future educational interventions may benefit from the inclusion of PD-focused communication strategies that address the bespoke needs of residents living with PD to help staff better manage communication difficulties as the disease progresses [35]. Such training could also play a crucial role in reducing the stigma faced by individuals with PD through the development of empathy among care staff, ultimately enhancing the quality of interactions and care [36].

While the findings in this review suggest evidence of promise for educational interventions, the discussion and conclusions drawn must be interpreted with caution. The scarcity of the available literature, coupled with the poor reporting quality and methodological limitations in the included studies, constrains the robustness of the evidence. These factors limit the strength of the conclusions that can be made and render critical discussion tentative in nature. Nevertheless, this review remains an important contribution as it synthesizes the limited evidence available and highlights the potential value of PD education and training in care home settings.

PD-specific training for care home staff has been recommended to enhance the care provided to people living with PD [10]. The included literature highlights the potential for such education to improve staff knowledge and, in turn, the experiences and care outcomes of those living with PD in care homes. However, the limited literature available suggests that PD education and training within care homes is an overlooked area. The small amount of literature available correlates with the finding of a low number of participants across the included studies having had previous training. Although it is possible that care providers offer training on PD to their staff but do not publish research regarding these training programs, it is not possible to assess the quality or effectiveness of such programs. The scarcity of the available literature represents a limitation of this ScR, as conclusions are drawn based on a limited body of literature and likely do not fully reflect the current state of educational and training interventions available to care home staff. This also highlights the need for further empirical research on PD education and training for staff within care homes.

The findings of this review should be seen as a starting point, providing a snapshot of what is currently available and guiding future work in this underexplored area. Despite these limitations, this synthesis is valuable in drawing attention to the need for more robust and well-reported studies, which could strengthen the conclusions and provide clearer insights into the impact of educational interventions for care home staff working with PD patients.

The included literature is also limited by a lack of follow-up, as, for example, one study faced challenges with low response rates to follow-up surveys [27]. Others did not provide detailed post-intervention knowledge assessments raising concerns about the long-term impact [23,29]. Therefore, future research may benefit from evaluating the long-term impacts of training interventions, such as knowledge retention and long-term improvements to care outcomes.

The reviewed literature suggests that training or educational interventions on PD may improve knowledge about PD and the care provided to people living with PD in care home settings. However, due to the scarce amount of available literature, future research should focus on the development, evaluation, and implementation of widespread PD education for care home staff.

### 4.1. Relevance to Nursing Practice

The findings of this scoping review have significant implications for nursing practices in care home settings, particularly in terms of the care of residents with PD. One of the key outcomes is the demonstrated effectiveness of educational interventions in enhancing care home nursing staff’s knowledge and confidence in managing PD. Within this review, structured PD-specific training programs were shown to significantly improve both perceived and actual knowledge levels among nursing staff, which, in turn, can lead to more effective symptom management, accurate medication administration, and overall better care for residents with PD. The sustained improvements in knowledge over time further suggest that ongoing education should be integrated into nursing practices as a routine part of professional development. This review also found that regularly updating nurses’ skills through such training not only builds upon existing knowledge but also ensures that the quality of PD care for residents continues to improve.

Another important implication is the apparent need for specialized training tailored to the complex and fluctuating nature of PD symptoms. Nursing practice should prioritize the development and implementation of comprehensive educational programs that address these unique challenges. Additionally, this review emphasizes the importance of communication training as a vital component of care for PD residents, many of whom experience speech and language difficulties. Nurses should be equipped with specific communication strategies, including alternative methods, to interact more effectively with residents. By incorporating these specialized communication skills into their practice, nurses can significantly improve their quality of care and enhance the overall quality of life for PD residents in care home settings.

### 4.2. Strengths and Limitations

This scoping review possesses several key strengths in its evaluation of educational and training interventions for care home staff working with Parkinson’s disease (PD) patients. A comprehensive search strategy was employed, utilizing six major electronic databases and following a well-defined approach based on the PEO framework. This ensured a thorough and systematic exploration of the relevant literature. The review adhered to robust methodological standards, aligning with established guidelines, such as the PRISMA-ScR checklist and Arksey and O’Malley’s framework for scoping reviews. Additionally, despite not being a mandatory component of scoping reviews, the authors conducted a quality appraisal of the included studies, which offers insights into the robustness of the evidence base. To the authors’ knowledge, this is also the first review of its kind. Despite these strengths, several limitations must be acknowledged. While the search strategy was rigorous, the review yielded only seven studies that met the inclusion criteria, reflecting the paucity of available research in this area. This small sample size may limit the generalizability of the findings and restrict the depth of analysis that could be conducted. Additionally, the heterogeneity of the included studies, in terms of methodologies, interventions, and outcome measures, posed challenges for the direct comparison and synthesis of results.

The quality appraisal revealed that none of the included studies were classified as high quality, with most deemed of moderate quality. This highlights an urgent need for more rigorous and methodologically robust research to strengthen the evidence base. Furthermore, six of the seven included studies utilized quantitative methodologies, which limited the availability of qualitative data, such as participant quotes, to provide richer contextual insights into the identified themes.

The scope of this review was restricted to studies published in English, which may have led to the exclusion of potentially relevant research in other languages. Finally, the exclusive focus on care home settings, while aligned with the review’s objectives, may limit the applicability of the findings to other care contexts for people with PD. This scoping review was also retrospectively registered, which represents a limitation as it introduces the potential for bias in study selection and reporting. Prospective registration would have ensured greater transparency in the review process and alignment with best practices for conducting systematic and scoping reviews.

We recognize that a more robust body of literature, including a broader range of high-quality studies and more diverse methodologies, would strengthen the conclusions and provide a more comprehensive understanding of this important topic. Future research should aim to address these gaps to better inform educational and training interventions for care home staff working with individuals with Parkinson’s disease.

## 5. Conclusions

In conclusion, while this scoping review highlights the potential benefits of training and educational interventions for care home staff managing Parkinson’s disease, it also highlights an important gap in the existing literature. The limited number of studies and the absence of published empirical research on these interventions suggest a need for further investigation. Future efforts should focus on developing and evaluating comprehensive, evidence-based educational programs that address the complex and fluctuating nature of PD symptoms. These programs should be tailored to the specific needs of care home staff and incorporate practical, hands-on training alongside theoretical knowledge. By addressing the identified gaps and building on the strengths of existing interventions, there is significant potential to enhance the quality of care for people with PD in care home settings

## Figures and Tables

**Figure 1 nursrep-15-00020-f001:**
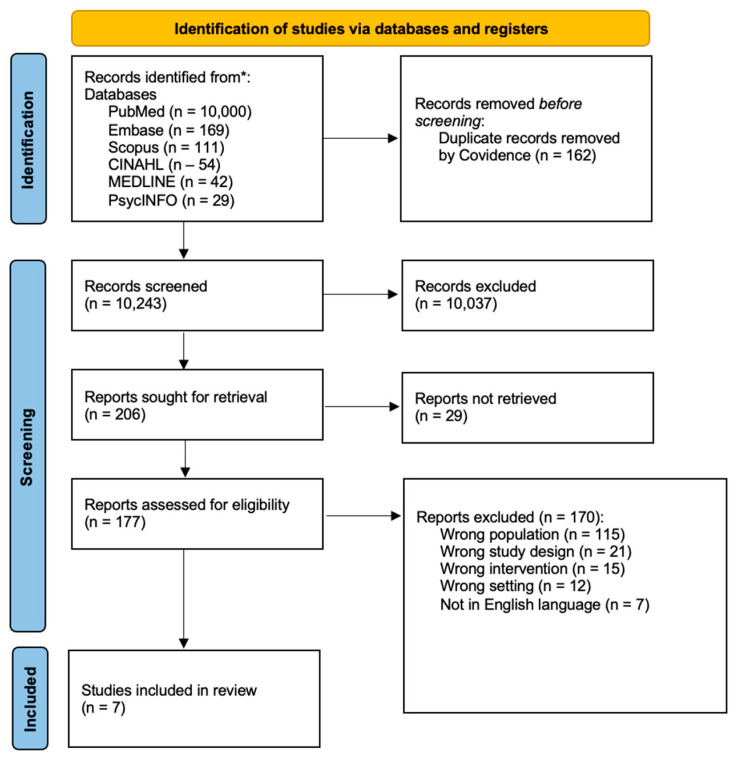
PRIMSA-ScR flowchart. * PRISMA: Preferred Reporting Items for Scoping Reviews and Meta-Analyses.

**Table 1 nursrep-15-00020-t001:** Search terms (PEO).

Population	Residential homes OR residential home OR Nursing homes OR nursing home OR Long term care facilities OR long-term care facility OR care home OR care homes OR aged care facility OR care home staff OR nursing home staff OR residential home staff
Exposure	Parkinsons OR Parkinson’s Disease OR Parkinsonian, Parkinsonian OR PD OR Progressive Supranuclear Palsy OR Multiple System Atrophy OR Corticobasal Degeneration OR Parkinson OR Movement Disorder
Outcome	Education OR Educational Activities OR Nursing Education OR Training OR Self-Directed Learning OR E-Learning OR Learning OR Staff Training OR CPD OR Continuing Professional Development OR educational intervention OR online education OR staff education OR staff development

**Table 2 nursrep-15-00020-t002:** Eligibility criteria.

	Inclusion	Exclusion
Study Characteristics	Empirical studies Literature reviews English language	
Population	Care home settings and staff (residential homes, nursing homes, long-term care facilities, aged care facilities, care homes)	Hospital settings, community settings other than care homes
Exposure	Parkinson’s disease, Parkinsonism, or movement disorders	
Outcome	Training, education, development, learning	

**Table 3 nursrep-15-00020-t003:** Sample information of included studies.

Authors	Title	Sample Size	Age	Gender	Professional Background
Ashraf et al., 2012 [23]	Impact of Care Home Education Program on the Care Home Staff and their Practice	15	Not provided	Not provided	Care home staff (not specified if RN or social care staff)
Chenoweth et al., 2013 [24]	Impact of the PD medication protocol program on nurses’ knowledge and management of PD medicines in acute and aged care settings	127	18–24—2% 25–34—17% 35–44—14% 45–54—35% 55+—32%	Female—95% Male—5%	RN: 67%; director of nursing: 1%; deputy director of nursing: 2%; clinical nurse specialist: 2%; nursing unit manager: 1%; enrolled nurse/trainee enrolled nurse: 2%; nurse assistant: 25%
Coles et al., 1995 [29]	Coping with communication disability in residential care	150	Not provided	Not provided	Social care staff
Eriksson et al., 2016 [28]	Communication partner training of enrolled nurses working in nursing homes with people with communication disorders caused by stroke or PD	1	45 years old	Female	Enrolled Nurse in nursing home with 5 months experience caring for a person with communication difficulties related to PD
Makoutonina et al., 2010 [25]	Optimizing care for residents with Parkinsonism in supervised facilities	118	Not provided	Not provided	Residential facility staff
Oates et al., 2016 [26]	Barriers to providing quality care for people with Parkinson’s living in care homes in the UK: The role of staff training	53 care home staff	Not provided	Not provided	Care home staff (not specified if RN or social care staff)
Wong and Luthra, 2019 [27]	Bridging the gaps in Parkinson’s education for nurses in long term care facilities	Not provided	Not provided	Not provided	Registered nurses in long-term care

**Table 4 nursrep-15-00020-t004:** Table of quality appraisal scores.

Study	Score
Ashraf et al. (2012) [23]	6
Chenoweth et al. (2013) [24]	7
Coles et al. (1995) [29]	4
Eriksson et al. (2016) [28]	7
Makoutonina et al. (2010) [25]	6
Oates et al. (2016) [26]	6
Wong and Luthra (2019) [27]	6

## Data Availability

Not applicable.

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
