# Peer review of "A Scoping Review of Educational and Training Interventions on Parkinson’s Disease for Staff in Care Home Settings"

_nursrep, 2025, doi:10.3390/nursrep15010020_

Round 1
Reviewer 1 Report
Comments and Suggestions for Authors
Dear Editor,
I appreciate the opportunity to review the article titled "A Scoping Review of Educational and Training Interventions on Parkinson’s Disease for Staff in Care Home Settings." The aim of this review was to synthesize evidence on PD education and training available to care home staff, examine existing programs and their effectiveness, and identify gaps in current educational approaches.
The topic is of paramount importance; however, I suggest the following reflections:
The choice of methodology: Does a scoping review allow for achieving the objectives of examining existing programs and their effectiveness and identifying gaps in current educational approaches? A systematic review might be more suitable.
Thematic analysis to identify key patterns and themes with only 7 studies?
With 7 studies, does it cover all these themes? How was this thematic analysis conducted? Thematic analysis revealed four main themes: improvements in PD knowledge and confidence, improvements in care practices and outcomes, the need for increased specialist education, and the incorporation of communication training.
This type of study does not permit the assertion that: "Conclusions: This review demonstrates the positive impact of PD-specific education on care home staff knowledge and practice."
Methodology: Data Analysis: Please explain and present these codes: This ScR employed thematic analysis, a robust method to identify, analyze, and interpret patterns within data [21]. This involved a six-step framework including familiarization with the data, generation of initial codes, searching for themes, reviewing themes, defining themes, and lastly reporting themes [22]. This process was led by SF, with rechecking, support, and guidance provided by the review team.
The figures presented in the PRISMA flow diagram are incorrect. The data presented from the article extraction are very limiting.
I consider that before proceeding to publication, the authors should review the methodological choices of the study and as a result, review the results and discussion.
Best Regards
Author Response
I appreciate the opportunity to review the article titled "A Scoping Review of Educational and Training Interventions on Parkinson’s Disease for Staff in Care Home Settings." The aim of this review was to synthesize evidence on PD education and training available to care home staff, examine existing programs and their effectiveness, and identify gaps in current educational approaches.
Thank you for taking the time to peer-review this submission,
The topic is of paramount importance; however, I suggest the following reflections:
The choice of methodology: Does a scoping review allow for achieving the objectives of examining existing programs and their effectiveness and identifying gaps in current educational approaches? A systematic review might be more suitable.
Thank you for this query. While a systematic review might would support in the assessment of intervention effectiveness, the scoping review methodology was chosen for several reasons:
- The broad nature of the research question, which aims to map the available evidence on PD education in care homes.
- The anticipated heterogeneity of the available literature, which included various study designs and outcomes.
- The exploratory nature of the review, which sought to identify gaps and future research directions.
We have added the following rationale to the manuscript: “scoping review methodology was selected as it was well-suited to the objectives of map-ping evidence, exploring existing programmes, and identifying gaps in PD education in care homes. The broad nature of the research question required a method capable of encompassing diverse study designs, outcomes, and approaches within the available literature. Anticipated heterogeneity in the evidence base further supported the choice of a scoping review over a systematic review. Additionally, the exploratory aim to identify gaps and guide future research aligned with the scoping review methodology. ”.
Thematic analysis to identify key patterns and themes with only 7 studies? With 7 studies, does it cover all these themes? How was this thematic analysis conducted? Thematic analysis revealed four main themes: improvements in PD knowledge and confidence, improvements in care practices and outcomes, the need for increased specialist education, and the incorporation of communication training.
Methodology: Data Analysis: Please explain and present these codes: This ScR employed thematic analysis, a robust method to identify, analyze, and interpret patterns within data [21]. This involved a six-step framework including familiarization with the data, generation of initial codes, searching for themes, reviewing themes, defining themes, and lastly reporting themes [22]. This process was led by SF, with rechecking, support, and guidance provided by the review team.
Thank you for these comments. We have provided greater detail in section 2.5 and rewritten our section on data analysis: “This scoping review employed thematic analysis as a systematic approach to identifying, analyzing, and interpreting patterns within the data [21]. The analysis followed a six-step framework as outlined by Braun and Clarke [22]. First, data familiarization was under-taken, where SF reviewed the extracted data multiple times to ensure a deep understand-ing of the content. This step involved reading and re-reading the data, making initial notes, and identifying potential areas of interest. Next, initial codes were generated through an inductive process, systematically labeling features of the data relevant to the research objectives. Codes were applied across the entire dataset to ensure comprehensive coverage. Following this, themes were searched by collating similar codes into broader categories that reflected meaningful patterns or concepts within the data. In the reviewing themes stage, these initial themes were critically evaluated to ensure coherence and alignment with the dataset. This involved refining themes by merging overlapping ones, discarding those that were not well-supported, and verifying their relevance to the re-search question. During the defining and naming themes step, the essence of each theme was clarified, with definitions established to distinguish them. Finally, the reporting of themes involved organizing the results into a coherent narrative supported by illustrative examples from the data. This iterative process was led by SF, with ongoing re-checking, guidance, and feedback provided by the review team to ensure rigor and reliability.”
This type of study does not permit the assertion that: "Conclusions: This review demonstrates the positive impact of PD-specific education on care home staff knowledge and practice."
Thank you for this helpful suggestion. We have amended accordingly: “This review provides evidence of promise regarding the potential impact of PD-specific education on care home staff knowledge and practice”.
The figures presented in the PRISMA flow diagram are incorrect. The data presented from the article extraction are very limiting.
Thank you for this. We have re-checked the PRISMA and confirm the figures are correct. We have however provided clarity on the process within the results: “A total of 10,405 studies were initially imported for screening. After removing 162 duplicates, 10,243 studies proceeded to title and abstract screening. Of these, 10,037 were deemed irrelevant and excluded. Full texts were sought for the remaining 206 studies; however, 29 studies could not be retrieved due to reasons such as inaccessible or outdated journal platforms, paywalls, or the studies not being available in any institutional or public database. 177 full-text studies were screened. Following this process, 7 studies met the inclusion criteria and were included in the review as illustrated in the PRISMA-ScR flowchart in Figure 1.”
Regarding data extraction, we appreciate your helpful comment and have provided further detail on this process: “For data extraction, a standardized extraction template was utilized within Covidence to ensure consistency across all studies. Key data points, including study design, population characteristics, intervention details, outcomes, and key findings, were extracted systematically. Both reviewers (SF and GM) independently performed data extraction for all included studies, with regular cross-checking to ensure accuracy. Any discrepancies in data extraction were discussed and resolved collaboratively to maintain reliability. “
I consider that before proceeding to publication, the authors should review the methodological choices of the study and as a result, review the results and discussion.
Thank you for your thoughtful feedback.
Scoping review justification: We have strengthened our rationale for choosing a scoping review methodology. This approach aligns well with our broad research question, which aims to map the available evidence on PD education in care homes, explore existing programs, and identify gaps in current approaches. The exploratory nature of our review and the anticipated heterogeneity in the literature further support this choice & we have now added this rationale within our manuscript. Thank you.
Adherence to PRISMA-ScR: Our review strictly follows the Preferred Reporting Items for Systematic Reviews and Meta-analysis extension for Scoping Reviews (PRISMA-ScR) guidelines. We have provided the completed PRISMA-ScR checklist as part of our submission to ensure transparency and adherence to best practices. We hope this provides confidence to the peer reviewers in our methods.
Coding and thematic analysis: We have added greater clarity regarding our coding and thematic analysis process. Our approach involved a six-step framework, including data familiarization, initial code generation, theme searching, theme reviewing, theme definition, and reporting. This process was led by the primary researcher with support and guidance from the review team.
Study selection process: We have provided more detailed information on how papers were selected for review, including our comprehensive search strategy across six electronic databases and the specific inclusion and exclusion criteria used.
We hope these clarifications address the queries while maintaining the integrity of our scoping review methodology. We thank you for your valuable input, which has helped us improve the quality and transparency of our manuscript.
Reviewer 2 Report
Comments and Suggestions for Authors
Dear authors,
Congratulations on completing this important review and submitting the manuscript for publication. I think the paper is generally well written and the methods are appropriate. However, I do have some concerns about the results and discussion sections.
First, I think that theme two is not adequately supported by the data. Only one study (25) was cited to show that PD training improves care and practice outcomes, specifically, reduced falls incidents. The second study (26) was quite vague and did not give any specific outcomes or practices that were improved. Can authors provide any further evidence from the review that support this theme?
Second, similar to theme two, I have the same same comment about theme four. The theme is not adequately backed by the findings from the reviewed studies. Can authors specify the number of training programs which had communication as part of the training module?
For theme three, it appears to be talking more about lack of specialist PD education for care home staff, and should probably be labelled as such and defined more appropriately. Authors appear to be interpreting the findings here, rather than reporting them. From lines 228 to 242, you cited numerous studies reporting low PD education among care home staff, including lack of knowledge and training in very critical areas. Clearly, lack of specialist training is the focus of this theme.
However, authors proceed to suggest that, based on the apparent lack of specialist education in many care home facilities, comprehensive specialist PD education programs should be organized to cover certain key areas. This is a recommendation based on the finding above, and should not become the name or focus of the theme.
DISCUSSION
Based on how the results (themes) were presented, the discussion followed a similar pattern, understandably so. Review of the discussion can be done only after the results section is revised.
Author Response
Dear authors,
Congratulations on completing this important review and submitting the manuscript for publication. I think the paper is generally well written and the methods are appropriate. However, I do have some concerns about the results and discussion sections.
Thank you for this supportive review and taking the time to review our manuscript.
First, I think that theme two is not adequately supported by the data. Only one study (25) was cited to show that PD training improves care and practice outcomes, specifically, reduced falls incidents. The second study (26) was quite vague and did not give any specific outcomes or practices that were improved. Can authors provide any further evidence from the review that support this theme?
Thank you for this feedback. We have included further data to support theme two as per the below:
Training programs led to measurable improvements in care outcomes. For instance, a falls prevention educational intervention in care homes, resulted in a decrease in the rate of falls by approximately 9% per month over the course of the 12-months following the in-tervention [25]. Another study reported that 64% of individuals with PD experienced neg-ative outcomes in respite or long-term care settings due to inadequate specialist training of care home staff but positive patient experiences were frequently associated with staff who had received thorough PD training in the form of improved quality of personalized PD care and greater adherence to medications [23].
Educational programs that focused on PD medication were commonly reported within this review [23-25]. In two Australian studies [24-25], education about PD medi-cations led to a variety of improvements in practice including increased adherence to medication schedules, better understanding of potential side effects, and improved man-agement of complex medication regimens. One study reported that their Parkinson’s Dis-ease Medication Protocol Program significantly increased nurses’ knowledge and confi-dence in managing PD medications, leading to more accurate medication administration and reduced errors [24]. Further, another study demonstrated that staff training on medi-cation management contributed to improved resident outcomes, such as better symptom control and reduced hospital admissions related to medication issues. These findings emphasize the importance of targeted medication education in enhancing both staff com-petency and resident care quality [25].
In addition to reduced falls, improvements in personalized care and medication ad-herence, education about PD can also support development of communication practice in nurses working within care homes [28]. In this Swedish study, a communication partner training program, between nurses and residents with PD, led to perceived improvements in quality of communication post-intervention from both staff and residents.
Second, similar to theme two, I have the same same comment about theme four. The theme is not adequately backed by the findings from the reviewed studies. Can authors specify the number of training programs which had communication as part of the training module?
Thank you for this feedback. We have rewritten the fourth theme as below:
“Communication was consistently identified as a critical component of PD care in educational interventions. Evidence from several studies demonstrated the benefits of communication training for care home staff, though the progressive nature of PD often limited the sustainability of these improvements. These programs consistently highlight-ed the vital role of communication skills in effective care, even when not specifically tai-lored to the distinct challenges posed by PD.
One study found that training care home staff as communication partners enhanced interactions with residents living with PD, though the long-term benefits were diminished due to the degenerative progression of the disease [28]. This highlights the necessity of communication strategies that adapt to the changing needs of individuals with PD.
Another study incorporated communication training into an educational program that included a module on PD, highlighting the importance of communication in deliver-ing quality care [29]. Similarly, a different program integrated modules on communication and PD, emphasizing the relevance of these skills in care settings [23]. However, neither program focused specifically on the unique communication challenges experienced by individuals with PD, suggesting an area for further development [23,29].
The use of communication partner training, a method previously validated for indi-viduals with aphasia, was evaluated in a study involving a nurse and a PD resident dyad [28]. While initial improvements in communication were observed, the effects were not sustained due to the resident’s progressive cognitive and motor decline.
Notably, two studies in this review developed their training programs broadly for care staff working with older adults rather than exclusively targeting PD-specific care [23,29]. These programs included general communication modules, which offered some indirect benefits for PD care. However, the programs did not address the distinct and pro-gressive communication challenges faced by people with PD, highlighting the need for more specialized training.”
For theme three, it appears to be talking more about lack of specialist PD education for care home staff, and should probably be labelled as such and defined more appropriately. Authors appear to be interpreting the findings here, rather than reporting them. From lines 228 to 242, you cited numerous studies reporting low PD education among care home staff, including lack of knowledge and training in very critical areas. Clearly, lack of specialist training is the focus of this theme. However, authors proceed to suggest that, based on the apparent lack of specialist education in many care home facilities, comprehensive specialist PD education programs should be organized to cover certain key areas. This is a recommendation based on the finding above, and should not become the name or focus of the theme.
Thank you for this helpful suggestion. We have amended the theme as suggested and rewritten the list of recommendations from studies as paragraphs to form a better narrative for the theme. The new theme is as follows: “3.3.3. Theme Three: Lack of Specialist PD Education
A recurring theme across the included studies was the lack of specialist PD education for care home staff. The included studies highlighted the low prevalence of previous PD-specific training among care home staff with studies reporting only 50% [29] and 59.6% [26] of care home staff having received prior PD-related training. Another study discovered that only 6.1% of aged care facility staff had received training or education specifically related to PD and its pharmacological management [24].
Significant gaps in nurses' knowledge regarding the management of PD, particularly in the administration of PD medications were also identified [24]. This study revealed that without specific clinical education, nurses often lack the necessary expertise to administer PD medications safely and effectively. The absence of such knowledge can result in seri-ous negative outcomes, such as delayed medication administration, dose omissions, or medication errors, which can adversely affect patient health and well-being [24].
Another study highlighted a significant gap in specialist training, specifically in the recognition and management of non-motor symptoms in individuals with PD [27]. De-spite the profound impact these non-motor symptoms can have on the quality of life for individuals with PD, none of the nurses in this study demonstrated awareness of non-motor symptoms prior to the educational intervention.
The included studies [24, 26, 27, 29] also provided recommendations about develop-ing specialist PD education in care homes. First, future educational programs should en-sure that staff are well-versed in the full spectrum of PD symptoms, including both motor and non-motor symptoms. Non-motor symptoms, which are often under-recognized and under-treated, require particular attention [27]. Additionally, specialized education on medication management is critical, as safe and effective medication practices are essential for preventing errors and improving patient outcomes [24]. The studies also emphasize the importance of ongoing training, with regular refresher courses and updates integrated into professional development to ensure staff stay informed about the latest advances in PD care and retain essential knowledge [24, 26]. Furthermore, expanding access to spe-cialized PD training programs is vital, especially given the low percentage of staff who have received prior PD education. This could include mandatory training for all care home staff, ensuring that educational resources are widely available and accessible to im-prove overall care quality [23, 26].”
DISCUSSION
Based on how the results (themes) were presented, the discussion followed a similar pattern, understandably so. Review of the discussion can be done only after the results section is revised.
Thank you for this supportive feedback which has helped us to develop our manuscript. We have retained our discussion on the basis of the development of themes 2, 3 and 4 as suggested in your review.
Round 2
Reviewer 1 Report
Comments and Suggestions for Authors
Thank you for the opportunity to review the article "A Scoping Review of Educational and Training Interventions on Parkinson's Disease for Staff in Care Home Settings."
I commend the authors for their work in conducting the review. However, for greater clarity of the data: Consider adding specific examples or direct quotes from the included studies to illustrate each identified theme. This could help reinforce the relevance and impact of the thematic findings.
Congratulations on the work developed.
Author Response
Thank you for the opportunity to review the article "A Scoping Review of Educational and Training Interventions on Parkinson's Disease for Staff in Care Home Settings."
I commend the authors for their work in conducting the review. However, for greater clarity of the data: Consider adding specific examples or direct quotes from the included studies to illustrate each identified theme. This could help reinforce the relevance and impact of the thematic findings.
Congratulations on the work developed.
Thank you for your supportive reviews. We have reviewed our themes and note that we have included data from all included papers referenced in our reviews. As 6 out of the 7 included studies are quantitative we are limited in the direct quotes we can provide. The further study we included in mixed methods, but there are limited quotations reported. As such, we have written the following within the limitations section of our manuscript to provide transparency:
“Despite these strengths, several limitations must be acknowledged. While the search strategy was rigorous, the review yielded only seven studies that met the inclusion criteria, reflecting the paucity of available research in this area. This small sample size may limit the generalizability of the findings and restrict the depth of analysis that could be con-ducted. Additionally, the heterogeneity of the included studies, in terms of methodologies, interventions, and outcome measures, posed challenges for direct comparison and syn-thesis of results.
The quality appraisal revealed that none of the included studies were classified as high quality, with most deemed of moderate quality. This highlights an urgent need for more rigorous and methodologically robust research to strengthen the evidence base. Fur-thermore, six of the seven included studies utilized quantitative methodologies, which limited the availability of qualitative data, such as participant quotes, to provide richer contextual insights into the identified themes.
The scope of this review was restricted to studies published in English, which may have led to the exclusion of potentially relevant research in other languages. Finally, the exclusive focus on care home settings, while aligned with the review's objectives, may limit the applicability of the findings to other care contexts for people with PD.
We recognize that a more robust body of literature, including a broader range of high-quality studies and more diverse methodologies, would strengthen the conclusions and provide a more comprehensive understanding of this important topic. Future re-search should aim to address these gaps to better inform educational and training inter-ventions for care home staff working with individuals with Parkinson's disease.”
Reviewer 2 Report
Comments and Suggestions for Authors
Dear authors,
Thank you for making efforts to revise your manuscript in line with the review suggestions. It is definitely an improvement over the initial draft. However, I think the quality of the review has been impacted by the small sample of included studies. Although you tried to shore up the evidence to support some of the themes, it is clear that you do not have much literature from the review to adequately support your themes. Therefore, I do not have any additional comments at this stage. Good luck with your research.
Author Response
Thank you for making efforts to revise your manuscript in line with the review suggestions. It is definitely an improvement over the initial draft. However, I think the quality of the review has been impacted by the small sample of included studies. Although you tried to shore up the evidence to support some of the themes, it is clear that you do not have much literature from the review to adequately support your themes. Therefore, I do not have any additional comments at this stage. Good luck with your research.
Thank you for this supportive review. To provide greater transparency in our work, we have added the following two paragraphs to our discussion:
1.While the findings in this review suggest evidence of promise for educational inter-ventions, the discussion and conclusions drawn must be interpreted with caution. The scarcity of available literature, coupled with poor reporting quality and methodological limitations in the included studies, constrains the robustness of the evidence. These fac-tors limit the strength of the conclusions that can be made and render critical discussion tentative in nature. Nevertheless, the review remains an important contribution as it syn-thesizes the limited evidence available and highlights the potential value of PD education and training in care home settings.
- The findings of this review should be seen as a starting point, providing a snapshot of what is currently available and guiding future work in this underexplored area. Despite the limitations, this synthesis is valuable in drawing attention to the need for more robust and well-reported studies, which could strengthen the conclusions and provide clearer insights into the impact of educational interventions for care home staff working with PD patients.
Further, we have rewritten the limitations section of our manuscript as follows:
Despite these strengths, several limitations must be acknowledged. While the search strategy was rigorous, the review yielded only seven studies that met the inclusion criteria, reflecting the paucity of available research in this area. This small sample size may limit the generalizability of the findings and restrict the depth of analysis that could be con-ducted. Additionally, the heterogeneity of the included studies, in terms of methodologies, interventions, and outcome measures, posed challenges for direct comparison and syn-thesis of results.
The quality appraisal revealed that none of the included studies were classified as high quality, with most deemed of moderate quality. This highlights an urgent need for more rigorous and methodologically robust research to strengthen the evidence base. Fur-thermore, six of the seven included studies utilized quantitative methodologies, which limited the availability of qualitative data, such as participant quotes, to provide richer contextual insights into the identified themes.
The scope of this review was restricted to studies published in English, which may have led to the exclusion of potentially relevant research in other languages. Finally, the exclusive focus on care home settings, while aligned with the review's objectives, may limit the applicability of the findings to other care contexts for people with PD.
We recognize that a more robust body of literature, including a broader range of high-quality studies and more diverse methodologies, would strengthen the conclusions and provide a more comprehensive understanding of this important topic. Future re-search should aim to address these gaps to better inform educational and training inter-ventions for care home staff working with individuals with Parkinson's disease